# Factors of prescribing phage therapy among UK healthcare professionals: Evidence from conjoint experiment and interviews

Sophie McCammon[1], Kirils Makarovs[2]*, Susan Banducci[3], Vicki Gold[1]

1 Living Systems Institute, Faculty of Health and Life Sciences, University of Exeter, Exeter, United Kingdom,
2 Faculty of Social and Behavioural Sciences, University of Amsterdam, Amsterdam, Netherlands,
3 Department of Social and Political Sciences, University of Exeter, Exeter, United Kingdom

* k.makarovs@uva.nl

**Data Availability Statement:** The research data supporting this publication are openly available at Figshare (https://doi.org/10.6084/m9.figshare.25435894).

## Abstract

With the global challenge of antimicrobial resistance (AMR), interest in the development of antibiotic alternatives has surged worldwide. While phage therapy is not a new phenomenon, technological and socio-economic factors have limited its implementation in the Western world. There is now a resurged effort, especially in the UK, to address these challenges. In this study, we collect survey data on UK general practitioners (n = 131) and other healthcare professionals (n = 103), as well as interviews with medical professionals (n = 4) and a focus group with medical students (n = 6) to explore factors associated with their willingness to prescribe phage therapy to patients. The interviews with medical professionals show support for the expansion of bacteriophage clinical trials and highlight their role as a viable alternative to antibiotics. A conjoint experiment reveals that success rate, side effect rate, and patient attitude to treatment are the decisive factors when it comes to phage therapy prescription; in contrast, the effects of administration route, type of treatment, and severity of infection were not statistically significant. Moreover, we show that general practitioners overall are more likely to recommend phage treatment to patients, compared to other healthcare professionals. The results of the study suggest that phage therapy has a potential to be widely accepted and used by healthcare workers in the UK.

## Introduction

Bacteriophages (or phages) are viruses that selectively infect specific target bacteria [1]. Phage therapy involves administering phages into a patient to eradicate bacterial infection [1]. Whilst phage therapy was first explored over a century ago [2, 3], its use in the Western world has been limited. This can be attributed to many factors including the discovery of antibiotics shortly following bacteriophage discovery and socio-political climate, which together resulted in phage-related research being isolated in Eastern Europe and India [4, 5].

Phage therapy is considered both safe and effective [6, 7]. However, there are challenges to phage therapy implementation in the UK, including the difficulty of accessing phages generated according to existing quality standards [6, 8, 9], insufficient private funding [9] and the

**Funding:** This research was funded by a Wellcome Trust Institutional Strategic Support Fund for Translational Research Exchange at Exeter (TREE), the Living Systems Institute, the Faculty of Health and Life Sciences, and the Faculty of Humanities and Social Sciences at the University of Exeter. The funders had no role in study design, data collection and analysis, decision to publish, or preparation of the manuscript.

**Competing interests:** The authors have declared that no competing interests exist.

complexity of clinical trials [10]. Together, these factors have confined usage to compassionate cases [11], conferring on phage therapy a perception of being somewhat experimental. Additionally, it is important to acknowledge the influence of public and political dynamics that can either support or obstruct the adoption of such treatments [8, 12].

Factors restricting phage therapy implementation in the UK include the classification of phages as medicinal products. This means phage therapy manufacturing and clinical studies must comply with both good manufacturing and clinical practices [6, 13, 14]. However, phages do not lend themselves easily to current development regulations imposed on other medicines such as antibiotics [8, 9]. Additionally, the complexities in patenting laws for biological products have resulted in minimal commercial interest in bacteriophage research [8].

However, the antimicrobial resistance (AMR) crisis has revitalised interest in phage therapy research; there is now a focus on addressing some of the associated challenges in the UK [6]. For example, the government's Science and Technology Select Committee currently has an open inquiry: "The antimicrobial potential of bacteriophages". This has multiple lines of inquiry, both on exploring the technological advances in this area of research as well as how policy changes could facilitate successful phage therapy implementation. Much of this has involved investigating how phage therapy has been successfully implemented internationally to anticipate challenges which may be faced and how to address these.

Investigation and anticipation of the potential socio-political factors which may hinder phage therapy implementation is vital. While there has been recent investigation into the UK public's awareness, acceptance, preferences and opinions regarding phage therapy, with an aim to provide insights into how phage treatment could be effectively integrated into society with the highest level of acceptance [15], the opinions of other integral groups remain elusive. A recent study conducted in Australia revealed most physicians surveyed (n = 92) believed phage therapy may be an effective solution to AMR, however, concerns surrounding ease of bacteriophage procurement was noted [16]. Phage therapy use in Australia is more established than in the UK due to amended regulatory laws to encourage bacteriophage research and development. Thus, how the opinions of Australian physicians compare to the UK, where phage therapy is in a much earlier stage of development and implementation, will be insightful.

Considering this, the aim of this study is to evaluate UK medical professionals' acceptance and concerns surrounding phage therapy implementation, and their prescribing preferences, which could help guide the development of phage products to maximise uptake rate. Our method of measurement is fielding a survey with an embedded conjoint experiment, also known as a discrete choice experiment (DCE). This was conducted with medical professionals with various healthcare backgrounds. This allows large-scale assessment of medical professional's acceptance of phage therapy and preferences regarding its implementation. One-to-one interviews with prescribing medical professionals also facilitated in depth discussion about their opinions surrounding AMR, phage therapy and other antibiotic-alternative treatments.

## Data and methods

This research consists of preliminary data gathering in the form of interviews, in addition to a survey fielded to two samples of UK medical professionals: general healthcare staff and general practitioners (GPs). From the survey, the main data reported is from an embedded DCE. Details of method design and data collection are provided in the following sections.

The study was approved by and adheres to the regulations of the University of Exeter Faculty of Health and Life Sciences Ethics Committee. Consent to participate in this research was informed by the provision of an approved participant information sheet and completion of an online or written consent form.

## Interviews and focus group

The interviews and focus group were intended to provide preliminary data to be used to guide the creation of the survey, as well as independent insight into medical professionals' views on the potential of phage therapy in the UK. Potential interviewees were initially identified through local contacts of members of the research group, and thereafter by snowball sampling. Initial contact with potential participants was by email, where an expression of interest led to the provision of a participant information sheet and written consent form. Four medical professionals who prescribe, or have prescribed, were recruited and interviewed between 12[th] October 2022 – 17[th] November 2022. The focus group with six participants recruited via an email invitation was conducted on 2[nd] November 2022 and participants were given Amazon vouchers as a thank you for their time.

Topic guides (see S1 and S2 Files) were used to guide the interviews and focus group to help us identify a set of salient considerations about what does, did, or would, influence participants' prescribing choices; this included regulatory factors and medicine attributes. Participants' views on AMR and how it could be addressed were also evaluated, along with their acceptance and views on phage therapy.

Interviews were conducted using Microsoft Teams, all led by one researcher (SM). Using the Microsoft Teams functions', the interviews were audio-recorded and transcribed. Transcripts were edited to correct errors and remove identifying information. Responses were thematically analysed, with a focus on identifying attributes of medical treatments which were influential on participants' prescription preferences; this information was used to guide creation of the survey.

## Survey

Qualtrics was used to design an online survey. The first section consisted of socio-demographic questions, including age, gender, and educational level. The participants' trust of various regulatory bodies, such as NICE and local formularies, was also assessed. The second section contained the DCE, which is described below. The final section evaluated what participants believed the main barriers to phage therapy implementation were and the degree of their support for phage therapy development. An open-response question asking for any further thoughts on the potential of phage therapy was also included. The order of the questions in each section was randomised to minimise any influence the ordering had on the respondent's answers.

## Discrete choice experiment

Discrete Choice Experiments (DCEs) are valuable tools in the healthcare sector for assessing how different attributes of medicines influence treatment preferences of patients and medical professionals [17–19]. The key advantage of employing the DCE experimental design, in contrast to traditional survey experiments, lies in its capacity to account for multiple attributes and discern their causal effects. Our strategy for determining the attributes and levels for the DCE drew on both the information gathered from the expert interviews, a focus group and a literature review. Together, these methods identified six attributes which were identified as being influential on medical professional's prescribing preferences and relevant to our research. Table 1 defines the attributes selected to be included in the survey's DCE, along with their associated levels.

Table 1 provided the basis for generating two theoretical treatments, which were presented as sets of attributes (Fig 1). By soliciting responses from participants to express their preferences, we assessed the impact of each attribute on their prescribing decisions. Once

**Table 1. List of selected attributes and levels, with definitions.**

| Attribute | Definition | Levels |
|---|---|---|
| Side effects | All medicines may have side effects, including nausea, headache and tiredness. Here, it is measured how many patients will get mild side-effects from the treatment. | • 1% (1 in 100) people using this therapy get side effects.<br>• 5% (5 in 100) people using this therapy get side effects.<br>• 10% (10 in 100) people using this therapy get side effects.<br>• 20% (20 in 100) people using this therapy get side effects. |
| Success rate of therapy | A medical treatment can fail to resolve an infection for many reasons, meaning the patient would have to receive another course. Success rate measures how many people will need no further treatment after the original course. | • 20% (20 in 100) people need no further treatment.<br>• 50% (50 in 100) people need no further treatment.<br>• 80% (80 in 100) people need no further treatment. |
| Administration route | Medicines can be administered to the patient in various ways, using different devices. | • Oral<br>• Intravenous<br>• Inhalation |
| Type of Treatment | Combinations of various phage-types along with other treatments can be taken. In this case, all the options are administered in an identical manner. | • One phage only<br>• Combination of different phages<br>• Phage plus antibiotic combination<br>• Antibiotics only |
| Patient attitude to treatment | Ultimately it is the patient's choice on whether they take a prescribed medicine. Their hesitance represents their resistance to taking the treatment during the consultation. | • No hesitance<br>• Some hesitance<br>• Extreme hesitance |
| Severity of infection | Patients can have infections of different severity, over different time periods. Chronic infections persist for a long time, while acute infections have a rapid onset and can clear quickly. | • Chronic<br>• Acute |

participants made a discrete choice regarding which treatment they would be more inclined to recommend to a patient, they were then asked to rate, on a scale from 1 to 10, their likelihood of using each treatment (1 = "Not at all likely" and 10 = "Very likely"). Considering these two sections together, we should capture both participants' "discrete preferences" and "attitudes" towards the treatments [20]. Each participant encountered five hypothetical choice sets, each consisting of a random selection of five out of the six attributes presented in Table 1.

## Study participants and period of the study

We fielded the online survey using two panels of participants: the first was a sample of participants identified as "Healthcare Professionals" (doctors, nurses, paramedics, pharmacists, psychologists, veterinarians) from the Prolific Research Platform and the second was general practitioners (GPs) from Panelbase. The sample of healthcare professionals included four doctors; however, due to the international classification of occupations used by Prolific, it is not possible to tell whether they are general practitioners or not. The Prolific survey was fielded from the 1st to 25th of February 2022, with the sample size of 103. The Panelbase survey was fielded to the GPs on the 21st-24th of March 2022, with the total sample size of 131.

In the subsample of general practitioners, 60.8% of the respondents identify as White British; 22.5% as Asian British, and 10.8% as Black British. Almost half of the general practitioners (41.2%) report residing in London and four-fifths are working full-time. The average age of GPs is 35.7; 63% of surveyed GPs are men and 37% are women. With regards to the qualification, 42% of GPs have undergone postgraduate specialty and general practice training; 40.5% of them are registered with the General Medicine Council, and 30.5% with the Health and Care Professions Council.

Please read the descriptions of two potential treatments for an infection.

Which of the two treatments would you personally prefer to recommend to treat an infection, presuming they are both a suitable option for the patient?

|  | Treatment 1 | Treatment 2 |
|---|---|---|
| Severity of infection | Acute | Chronic |
| Administration route | Oral | Oral |
| Success rate | 80% (80 out of 100 people need no further treatment) | 50% (50 out of 100 people need no further treatment) |
| Type of treatment | Phage cocktail (mixture of different phages) | Phage plus antibiotic combination |
| Side effect rate | 10% (10 in 100) people using this therapy get side effects | 10% (10 in 100) people using this therapy get side effects |

I prefer Treatment 1

I prefer Treatment 2

On a scale from 1 to 10, where 1 indicates 'not at all likely' and 10 indicates 'very likely', what is your likelihood of recommending **Treatment 1**?

Not at all likely    Very likely
0    1    2    3    4    5    6    7    8    9    10

Treatment 1

Using the same scale, what is your likelihood of recommending **Treatment 2**?

Not at all likely    Very likely
0    1    2    3    4    5    6    7    8    9    10

Treatment 2

**Fig 1. Conjoint experiment: Example of screen seen by study participants.** Attribute levels were randomly assigned to create two hypothetical treatments. Participants were asked to express a preference for Treatment 1 or Treatment 2 and rank the likelihood of use of each treatment on a scale of 1 (Not at all likely) to 10 (Very likely).

In contrast to the subsample of GPs, the Prolific subsample of health professionals is predominantly comprised of white British (86.1%); geographically, they tend to be distributed across various regions of the United Kingdom. The overwhelming majority of surveyed Prolific health professionals are women (91%); the average age (44.5 years) tends to be somewhat higher than that of the GP subsample. Slightly more than half of health professionals report working full-time (55.4%); two-thirds (61.2%) report obtaining a university degree or higher; and 28.2% are not registered with any of the councils.

To maximise the power of this research, the two samples were combined to give an overall sample size of 234. Respondents who failed the attention test within the survey were discarded (n = 39) to preserve data integrity. This left an effective sample size of 195. For further sociodemographic information, refer to the data in S1 Table.

## Statistical analysis

In our DCE analysis, we utilised the cregg package developed by Leeper [21]. This allowed us to compute both the average marginal component effects (AMCE) and the marginal means. The AMCE can be seen as coefficients representing the "causal effect," while the marginal

means indicate the overall favourability of an attribute with the mean support ranging from 0 to 1. By examining marginal means, we gain descriptive insights into the attributes within our sample and understand the average outcome of each attribute. A mean above the midpoint indicates a positive effect on infection treatment preference, whereas a mean below the midpoint suggests a negative effect. Notably, marginal means are the preferred method for comparing subgroup differences given AMCE's sensitivity to baseline selection.

## Results

### Interviews & focus groups

A dentist, two former GPs who still worked in the medical sector and an active consultant in respiratory medicine were interviewed. All participants highlighted the importance of the NICE and local formulary guidelines in influencing their prescribing decisions. Of the attributes of the medicine itself, the medical professionals agreed that potential side effects, success rate and ease of administration all influenced their prescribing decisions, along with the cost of the treatment and any accompanying care. One participant stated that they considered the potential environmental impact of the medicine or administrative devices when prescribing.

All interviewees were aware of AMR, with three of the four having encountered AMR in their career. The dentist highlighted that due to their relatively low prescribing rate, AMR may not be as much of an issue as in other medical environments, such as hospitals. The interviewees also noted a change in the attitude towards AMR as time has passed, with increasing emphasis on antibiotic stewardship. However, they highlighted barriers to antibiotic stewardship that remain when an effective treatment is needed quickly, so a broad-spectrum antibiotic is usually the only option. All interviewees also noted the pressure from patients to prescribe antibiotics, some on a daily basis. While some explained their approach to address this was counselling and educating about AMR, the time pressures of the appointment often meant this was usually unachievable. Prescriptions of non-antibiotic treatments, such as antiseptic creams and paracetamol, were also used to alleviate the pressure to prescribe antibiotics.

Three of the four medical professionals had heard of phage therapy before the interview, with one being involved in phage research and prescription for the past eight years. All interviewees believed phage therapy was a viable alternative to antibiotics in the future; however, the potential societal and technological challenges were noted. All interviewees supported the expansion of bacteriophage clinical trials and suggested a combination of antibiotics and phages may ease societal hesitance. Preliminary use of phages to treat chronic infections, opposed to acute, was also suggested. This would alleviate issues regarding phage specificity as for serious long-term infections the infecting bacteria is often already known, making it easier to identify which phage would be effective. The interviewees also believed increased education on phage therapy and AMR was vital for successful implementation.

The focus group conducted with medical students at a university in the UK revealed similar considerations. Of the focus group participants, most had heard of phage therapy and all were aware of AMR. However, participants shared that their awareness of phage therapy was limited before the focus group. Some had encountered the term in their studies but didn't have in-depth knowledge. Some felt that AMR was a significant issue, while others believed it wasn't yet a critical problem and that other global issues might be more impactful.

In the discussion on phage therapy, the focus group identified three key attributes that significantly influence the prescribing decisions of medical professionals. The success rate of phage therapy was unanimously considered the most influential factor. The magnitude of treatment benefit was highlighted as a critical factor. The participants recognized that the extent of improvement in a patient's condition achieved through phage therapy would strongly

impact their decision-making. A treatment that could provide a substantial and clinically significant benefit would be more appealing to doctors when considering whether to prescribe it to their patients. These three attributes—success rate, time until treatment benefit, and magnitude of treatment benefit—were collectively recognized as the most influential factors guiding doctors' decisions regarding phage therapy prescriptions. The time until treatment benefit was deemed crucial, and the medical students identified the importance of a treatment's rapid onset of action, as patients often seek quick relief from infections. They suggested that the faster phage therapy could demonstrate its efficacy compared to conventional antibiotics, the more likely it would be considered as a treatment option.

## Survey

Only six participants indicated they were not aware of AMR; on a scale of 1–5 (1 = "Not concerned at all", 5 = "Extremely concerned") the mean level of concern for AMR was 3.93. Moreover, 83 (43.7%) individuals in the combined sample indicated they had heard of phage therapy before completing the survey; however, the proportion of awareness varied drastically between the general health professional sample and GP sample. For example, 81.7% of the GP sample had previously heard of phage therapy, whereas this was only 17.5% for the general health professionals' sample. In addition to some possible overreporting of awareness, these differences are likely due to GPs having higher qualification and being more experienced in treating infections, prescribing medicines and thus being aware of the strategies for managing AMR.

## Discrete Choice Experiment (DCE)

Fig 2 illustrates the direction and degree of influence the assessed attributes have on the participant's prescribing preferences. Success rate was the most influential attribute when deciding which treatment to recommend to a patient, followed by side effect rate. For example, increasing the success rate from 20% of people needing no further treatment to 80% increases the participants' mean likelihood of recommending the treatment to a patient by 0.29. Additionally, moving from a 20% side effect rate to a 1% side effect rate increases mean treatment preference by 0.20. Moreover, the level of patient hesitance towards the treatment also slightly influenced prescribing preferences; unsurprisingly, extreme patient hesitance had a negative influence on participants' likelihood of prescribing the treatment. The effects of severity of infection, administration route, and phage combination with antibiotics on treatment preference were not statistically significant (see S2 Table).

In addition to indicating which treatment they would prefer to prescribe out of the two hypothetical choices, medical professionals were asked to evaluate, on a scale from 1 to 10, their likelihood of recommending each hypothetical treatment to patients (Fig 3). This continuous analysis validates the results from the DCE above.

We then compared the responses given by the GP sample and general health professional sample (Fig 4). While there appears to be no significant difference between the two samples in the discrete choice analysis, there is an apparent difference in the preference rankings between the two samples. For almost all the attributes and their levels, GPs show a higher willingness to hypothetically recommend treatments to patients, compared to general health professionals.

In the survey of health professionals, we also asked an open-ended question eliciting any additional thoughts on phage therapy. The wording of the question was: "We would be interested in hearing any additional thoughts you have on phage therapy and whether it could provide an alternative to antibiotics in treating infections." Health professionals' responses regarding phage therapy as a response to the AMR crisis exhibit a mixture of optimism,

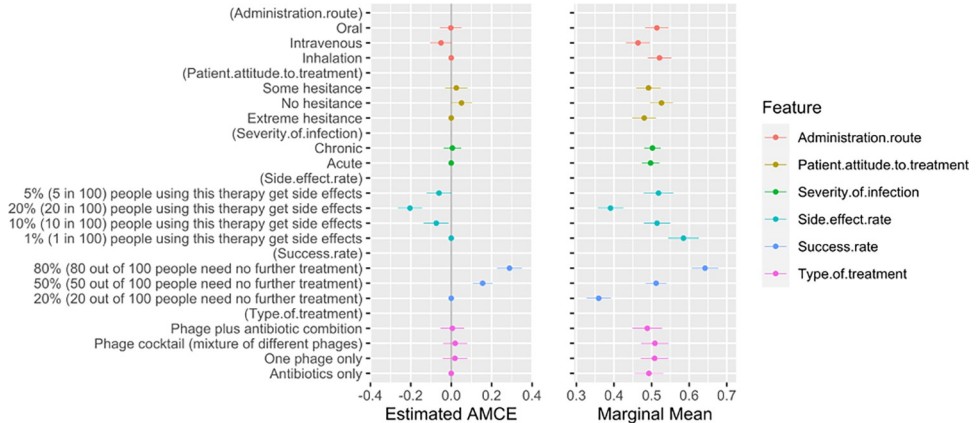

**Fig 2. AMCE & marginal means for discrete preferences.** The left panel shows the AMCE for each attribute level, while the right panel plots the marginal means. The AMCE represents the average conditional impact on prescription preference for each attribute compared to the baseline. On the other hand, the marginal mean reflects the overall favourability of an attribute on a scale from 0 to 1, where values above 0.5 signify a positive effect on treatment preference, and values below 0.5 suggest a negative effect. Whiskers represent 95% confidence intervals; significant associations have no overlap.

curiosity, and concerns. Many participants acknowledged a lack of in-depth knowledge about phage therapy but expressed a keen interest in learning more about this emerging alternative to antibiotics. They viewed it as a promising avenue, provided that rigorous scientific research and extensive clinical trials support its safety and efficacy. However, a recurring concern among respondents was the potential hesitancy of the general public towards phage therapy, emphasising the need for educational campaigns to instil confidence. Specific worries revolved around the possibility of heightened side effects, the practicality of administration methods (e.g. intravenous therapy), and the need for further evidence to address safety issues. Some respondents stressed the importance of understanding bacterial resistance to phages and the unpredictability of their interactions within the human body. Cost considerations were also

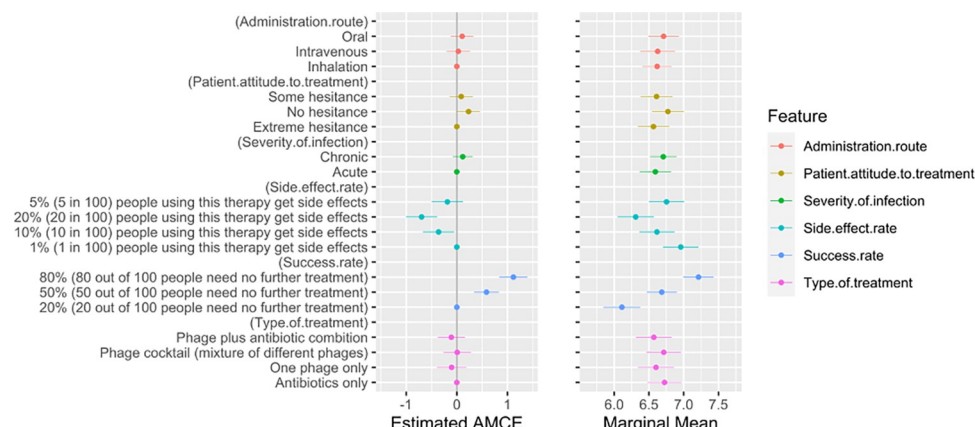

**Fig 3. AMCE and marginal means for preference rankings.** The left panel shows the AMCE for each attribute level, while the right panel plots the marginal means. The AMCE represents the average conditional impact on prescription preference for each attribute compared to the baseline. On the other hand, the marginal mean reflects the overall favourability of an attribute on a scale from 0 to 1, where values above 0.5 signify a positive effect on treatment preference, and values below 0.5 suggest a negative effect. Whiskers represent 95% confidence intervals; significant associations have no overlap.

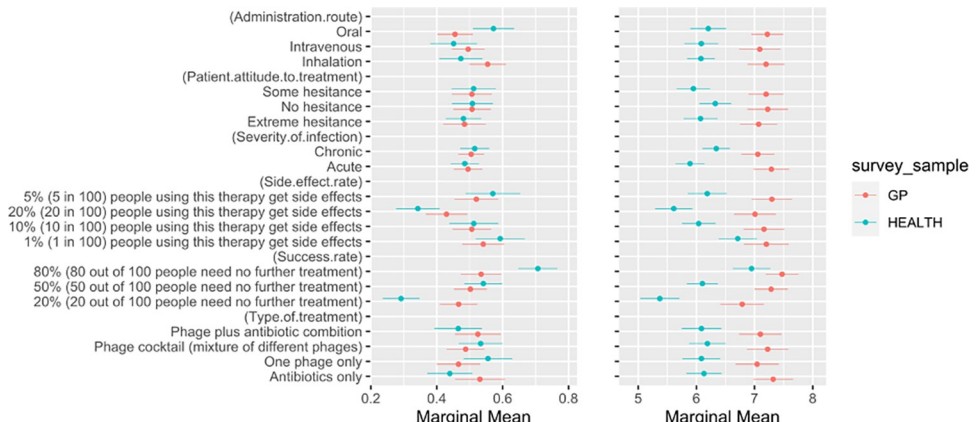

**Fig 4. Comparison of general health professional vs. GP preference rankings.** The left panel shows the marginal means for the participants' discrete treatment choices, split by sample; general health professionals (blue) and GPs (red). The right panel shows the marginal means for the participants' ranked preferences, again split by sample. Values above 0.5 signify a positive effect on treatment preference, and values below 0.5 suggest a negative influence on treatment preference. Whiskers represent 95% confidence intervals; significant associations have no overlap.

raised, with suggestions that any initial expenses might be offset by reducing antibiotic prescriptions in the long run.

The prevailing sentiment was a call for more research and clinical trials to provide the necessary data to support the widespread implementation of phage therapy. Additionally, some participants highlighted the need for increased awareness and regulation in this relatively unknown field.

## Discussion

The momentum surrounding phage therapy in the early 20th century slowed due to the rapid development of antibiotics. In nations like the UK, the present clinical application of phage therapy is primarily limited to specific cases where conventional treatments have proven ineffective. The involvement of commercial players in this field is limited, and comprehensive large-scale clinical trials are infrequent. Nevertheless, the escalating issue of AMR has sparked renewed international fascination with the possibilities offered by phage therapy [6, 8, 22] and there is a renewed effort to address these challenges. Public acceptance of a treatment greatly influences its success and uptake; our previous research shows an apparent support and acceptance of phage therapy amongst the UK public [15]. However, to our knowledge, opinions of UK medical professionals, who will be key players in dictating the success of phage therapy uptake, have yet to be assessed.

Congruent to the UK public, this study suggests there is a high level of support for the development and expanded use of phage therapy from UK medical professionals. As expected, this medical professionals' sample showed higher awareness of AMR and phage therapy than the lay UK public sample previously assessed [15]. However, the level of awareness about phage therapy differs drastically between GPs and other medical professions, with the former having a far greater awareness. This may reflect the difference in education syllabuses and day-to-day experiences of the two cohorts. Our evidence from the interviews and focus group also suggests that there is a great deal of interest in investing in both developing phage therapy as an alternative treatment, raising awareness in the public about the treatment and destigmatising viruses.

The experimental evidence suggests that, when deciding to prescribe phage therapy to patients, healthcare professionals consider the success and side effect rates to be the most decisive factors. This is congruent to how the general public justifies their preference towards phage therapy [15]; it also goes in line with the factors that are commonly related to antibiotic prescription [23]. This suggests that the phage therapy has a potential to be widely adopted as a means of substituting antibiotics and thus tackling AMR, should it be approved within the existing regulatory frameworks in the UK.

Another noticeable finding from the experiment is the consistently higher willingness of GPs to recommend phage treatment to patients, compared to general health professionals. The explanation for this could lie in the fact that GPs tend to have more hands-on experience in diagnosing and treating various medical conditions; they are also more likely to encounter patients who did not respond well to traditional antibiotics and thus are willing to try out alternative types of treatment. Further research in this area should focus on exploring the motivation of GPs to implement phage therapy by considering such factors as interaction with patients, continuing education, and personal interest.

It is important to acknowledge certain limitations in the conjoint experiment. Firstly, the number of attributes presented to respondents had to be limited to six or seven, with the number of corresponding attribute levels being restricted to three or four. Beyond this limit, participants may experience an increased cognitive burden, leading to shortcuts in their evaluation and decision-making processes [24]. Additionally, including more attributes reduces the power of the study and with the relatively small sample size, this was something we had to consider. Hence, there are other attributes of medicines (e.g. the drug development and approval process; peer effect; recommendations from professional associations) which may be highly influential on the participants' prescribing decisions which were not assessed in this study. Another constraint of the DCE experimental design is that it can only assess behaviours and attitudes that can be operationalized through discrete binary choice or ranking questions [25]. Finally, survey experiments, including DCEs, have been criticised for their artificial nature, potentially limiting their ability to fully represent real-world behaviours [26]. However, research by Hainmueller et al. suggests that the results obtained from conjoint experiments can closely approximate real-world behavioural benchmarks [27].

## Conclusion

The present research examined factors associated with the willingness of the UK healthcare professionals to prescribe phage therapy to patients. Overall, we found that there is a great deal of interest in developing and implementing phage therapy as an alternative type of treatment; we have also found that such factors as success rate, side effect rate, and patient attitude to treatment are the decisive ones when it comes to phage therapy prescription. Additionally, our findings show that general practitioners (GPs) tend to be more likely to prescribe phage therapy to patients, compared to other healthcare professionals. Future research should focus on the cross-cultural comparison of attitudes to phage therapy among healthcare workers and identify approaches to tackling barriers related to its usage in the UK and worldwide.

## Supporting information

**S1 File. Interview topic guide.**
(DOCX)

**S2 File. Focus group topic guide.**
(DOCX)

**S1 Table. Table of sample sociodemographic characteristics.**
(DOCX)

**S2 Table. Tables of coefficient estimates for the conjoint experiment.**
(DOCX)

## Author Contributions

**Conceptualization:** Sophie McCammon, Kirils Makarovs, Susan Banducci, Vicki Gold.

**Data curation:** Susan Banducci.

**Formal analysis:** Kirils Makarovs, Susan Banducci.

**Investigation:** Sophie McCammon, Kirils Makarovs, Susan Banducci, Vicki Gold.

**Methodology:** Kirils Makarovs, Susan Banducci.

**Project administration:** Sophie McCammon, Vicki Gold.

**Supervision:** Susan Banducci, Vicki Gold.

**Visualization:** Sophie McCammon, Kirils Makarovs, Susan Banducci.

**Writing – original draft:** Sophie McCammon, Kirils Makarovs, Susan Banducci.

**Writing – review & editing:** Sophie McCammon, Kirils Makarovs, Susan Banducci, Vicki Gold.

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
