## [Decision Letter · Decision Letter 0]

30 Jan 2024

PONE-D-23-33160Factors of Prescribing Phage Therapy among UK Healthcare Professionals: Evidence from Conjoint Experiment and InterviewsPLOS ONE

Dear Dr. Makarovs,

Thank you for submitting your manuscript to PLOS ONE. After careful consideration, we feel that it has merit but does not fully meet PLOS ONE’s publication criteria as it currently stands. Therefore, we invite you to submit a revised version of the manuscript that addresses the points raised during the review process.

Discuss the lack of randomness of the sample and the implications of respondents reside in London, being not representative of the rest of the country.

We look forward to receiving your revised manuscript.

Kind regards,

Adelaide Almeida

Academic Editor

PLOS ONE

Journal Requirements:

4. Please amend your list of authors on the manuscript to ensure that each author is linked to an affiliation. Authors’ affiliations should reflect the institution where the work was done (if authors moved subsequently, you can also list the new affiliation stating “current affiliation:….” as necessary)

Reviewers' comments:

Reviewer's Responses to Questions

**Comments to the Author**

1. Is the manuscript technically sound, and do the data support the conclusions?

Reviewer #1: Yes

Reviewer #2: Yes

2. Has the statistical analysis been performed appropriately and rigorously? 

Reviewer #1: Yes

Reviewer #2: Yes

3. Have the authors made all data underlying the findings in their manuscript fully available?

Reviewer #1: Yes

Reviewer #2: Yes

4. Is the manuscript presented in an intelligible fashion and written in standard English?

Reviewer #1: Yes

Reviewer #2: Yes

5. Review Comments to the Author

Reviewer #1: The manuscript "Factors of prescribing phage therapy among UK healthcare proffesionals: evidence from conjoint experiment and interviews" reletes to a topic which is frequently underestimated with regard to phage therapy namely perception of this form of threatment by peaple whose work is associated with medical care. The manuscript is well written, the results are clearly presented on the graphs as well as discussed in the text. The weakest fragment is discussion section. I think that it might be a little bit extended and a short paragraph as a conclusion should be added as the last fragment in this section. Nevertheless, considering the fact that the Authors presented non-nobvious and at the same time important aspect of phage treatment I think that the manuscript is worth publishing.

Reviewer #2: The authors focus on phage therapy perception among UK healthcare professionals. As phage therapy grows in popularity, there is an increasing need for analysis of public and health care professionals opinion regarding experimental therapies (these days phage therapy can be implemented solely on an experimental basis). The data presented support the conclusions (figures and supplementary materials) and DCE analysis (employed by the authors) have been increasingly used in health and health care settings. The sample size could be larger as 131 general practitioners and 103 healthcare professionals may not be a representative sample (more about this issue later in my comments). The English is fully understandable even for non-native speakers (like me). My other comments and concerns are as follows:

1. "Current factors restricting phage therapy implementation in the UK include phage classification as a medicinal product" - in my opinion the current factors restricting phage therapy implementation globally (not only in UK) is the lack of clinically confirmed efficacy (including successfully completed clinical trials). I think the experimental status of phage treatment needs to be emphasized more.

2. “Potential interviewees were initially identified through local contacts of members of the research group, and thereafter by snowball sampling” – I wonder how such identification of potential respondents (no random selection) could impact the results?

3. “Interviews were conducted using Microsoft Teams” – it's no secret that lack of anonymity can affect survey results (the respondent tries to please the interviewer). Comments from the authors on this issue would be welcome.

4. “Almost half of the general practitioners (41.2%) report residing in London and four-fifths are working full-time” – at this point lack of randomness is obvious but there is also another issue inevitably associated with such proportion of respondents. London is certainly not representative of the rest of the country. It is the opposite – the differences between London and the rest of UK are more than clear in almost every economic and social aspect. The authors did not mention about this issue at all and in my opinion it is crucial to at least highlight the possible differences.

5. “subsample of health professionals is predominantly comprised of white British (86.1%); geographically, they tend to be distributed across various regions of the United Kingdom.” - The authors should try to explain why there are such differences in geographical distribution between health professionals and GP.

6. “For example, 81.7% of the GP sample had previously heard of phage therapy, whereas this was only 17.5% for the general health professionals’ sample.” – this is exactly what I emphasized in my comment No. 4. GPs are better educated because half of them live in London. This geographical factor must be somehow addressed by the authors.

7. “However, the level of awareness about phage therapy differs drastically between GPs and other medical professions, with the former having a far greater awareness. This may reflect the difference in education syllabuses and day-to-day experiences of the two cohorts.” – such statement is not valid without analyzing health professionals who live in London. Education is only part of the problem. Again – London – is another important factor.

8. “Another noticeable finding from the experiment is the consistently higher willingness of GPs to recommend phage treatment to patients, compared to general health professionals. The explanation for this could lie in the fact that GPs tend to have more hands-on experience in diagnosing and treating various medical conditions” – again – the "London" issue/impact.

6. PLOS authors have the option to publish the peer review history of their article (what does this mean?). If published, this will include your full peer review and any attached files.

Reviewer #1: No

Reviewer #2: No

---

## [Author Response · Author response to Decision Letter 0]

19 Mar 2024

Dear Reviewers,

Thank you for your comments on the “Factors of Prescribing Phage Therapy among UK Healthcare Professionals: Evidence from Conjoint Experiment and Interviews” manuscript. We appreciate the time and effort that you dedicated to providing feedback on our manuscript and are grateful for the insightful comments. We have incorporated most of the suggestions into the manuscript. Please see below for a point-by-point response to reviewers’ comments.

Journal Requirements

The file names and the format for the figures have been updated, now they are in .tif format. The manuscript is now submitted in double-space paragraph format. The supplementary materials are now cited correctly and uploaded as separate documents. The “Supporting Information” section is added at the end of the manuscript.

2. Data Deposit

The data is available at Figshare. DOI: 10.6084/m9.figshare.25435894

This research did not receive any external funding and therefore no award number exists. The following information has been added to the Acknowledgements section:

This research was supported by a Wellcome Trust Institutional Strategic Support Fund for Translational Research Exchange at Exeter (TREE), the Living Systems Institute, the Faculty of Health and Life Sciences, and the Faculty of Humanities and Social Sciences at the University of Exeter.

4. Please amend your list of authors on the manuscript to ensure that each author is linked to an affiliation. Authors’ affiliations should reflect the institution where the work was done (if authors moved subsequently, you can also list the new affiliation stating “current affiliation:….” as necessary)

The affiliations of the authors have been updated.

Done.

The references have been reviewed and - to the best of our knowledge - are complete and correct.

Reviewers' comments:

Reviewer #1: The manuscript "Factors of prescribing phage therapy among UK healthcare proffesionals: evidence from conjoint experiment and interviews" reletes to a topic which is frequently underestimated with regard to phage therapy namely perception of this form of threatment by peaple whose work is associated with medical care. The manuscript is well written, the results are clearly presented on the graphs as well as discussed in the text. The weakest fragment is discussion section. I think that it might be a little bit extended and a short paragraph as a conclusion should be added as the last fragment in this section. Nevertheless, considering the fact that the Authors presented non-nobvious and at the same time important aspect of phage treatment I think that the manuscript is worth publishing.

Thank you for your comment. A concluding paragraph has been added to the manuscript to summarise the main findings.

Reviewer #2: The authors focus on phage therapy perception among UK healthcare professionals. As phage therapy grows in popularity, there is an increasing need for analysis of public and health care professionals opinion regarding experimental therapies (these days phage therapy can be implemented solely on an experimental basis). The data presented support the conclusions (figures and supplementary materials) and DCE analysis (employed by the authors) have been increasingly used in health and health care settings. The sample size could be larger as 131 general practitioners and 103 healthcare professionals may not be a representative sample (more about this issue later in my comments). The English is fully understandable even for non-native speakers (like me). My other comments and concerns are as follows:

1. "Current factors restricting phage therapy implementation in the UK include phage classification as a medicinal product" - in my opinion the current factors restricting phage therapy implementation globally (not only in UK) is the lack of clinically confirmed efficacy (including successfully completed clinical trials). I think the experimental status of phage treatment needs to be emphasized more.

Thank you for raising this point. We have added a comment regarding the experimental status of phage therapy, and for balance also cite recent work “The Future of Clinical Phage Therapy in the United Kingdom” by Jones et al, 2023 and “Safety and efficacy of phage therapy in difficult-to-treat infections: a systematic review” by Uyttebroek et al, 2022 that supports clinical efficacy of phage therapy.

The following paragraph has been added:

Phage therapy has been demonstrated as both safe and effective (Jones et al, 2023; Uyttebroek et al, 2022). However, there are challenges to phage therapy implementation in the UK, including the difficulty of accessing phages generated according to existing quality standards (Brives et al. 2020; Debarbieux et al. 2015; Jones et al, 2023), insufficient private funding (Debarbieux et al. 2015) and the complexity of clinical trials (Reindel et al, 2017). Together, these factors have confined usage to compassionate cases (McCalin et al. 2019) conferring on phage therapy a perception of being somewhat experimental. Additionally, it is important to acknowledge the influence of public and political dynamics that can either support or obstruct the adoption of such treatments (Brives et al. 2020; Hinchliffe et al. 2018).

Jones, J. D., Trippett, C., Suleman, M., Clokie, M. R., & Clark, J. R. (2023). The Future of Clinical Phage Therapy in the United Kingdom. Viruses, 15(3), 721.

Uyttebroek, S., et al. (2022). Safety and efficacy of phage therapy in difficult-to-treat infections: a systematic review. Lancet Infet Dis, 22(8):e208-e220.

Brives C, Pourraz J. (2020). Phage therapy as a potential solution in the fight against AMR: obstacles and possible futures. Palgrave Commun 6, 100.

Debarbieux L, Pirnay JP, Verbeken G, De Vos D, Merabishvili M, Huys I, et al. A bacteriophage journey at the European medicines agency. FEMS Microbiol Lett. 2015;363(2):2015–6. pmid:26656541

Reindel R, Fiore CR. (2017). Phage therapy: Considerations and challenges for development. Clin Infect Dis. 64(11):1589–90

McCallin S, Sacher JC, Zheng J, Chan BK. Current state of compassionate phage therapy. Viruses. 2019;11(4):1–14. pmid:31013833

Hinchliffe S, Butcher A, Rahman MM (2018). The AMR problem: demanding economies, biological margins, and co-producing alternative strategies. Palgrave Commun. 4, 142.

2. “Potential interviewees were initially identified through local contacts of members of the research group, and thereafter by snowball sampling” – I wonder how such identification of potential respondents (no random selection) could impact the results?

Thank you for this comment. We have used the snowball sampling for the qualitative part of the project, namely the interviews with the medical professionals. While indeed snowball sampling may end in a non-representative sample of healthcare professionals, this is not an issue for the qualitative research as it was more concerned with understanding healthcare professionals’ experience and opinions regarding phage therapy and generating relevant categories and attributes that would then form the basis for the conjoint experiment.

3. “Interviews were conducted using Microsoft Teams” – it's no secret that lack of anonymity can affect survey results (the respondent tries to please the interviewer). Comments from the authors on this issue would be welcome.

Microsoft Teams has been used for conducting qualitative interviews (Interviews and Focus Groups p 3) where interviewees are not anonymous to the researcher. Both surveys conducted within this research project were operated via Qualtrics and the anonymity of the respondents has been preserved. The possibility for change in responses by participants due to a social desirability effect noted by the reviewer would be of concern if the questions were of a sensitive nature but we do not consider questions to be of a sensitive nature. We may expect overreporting on awareness of phage therapy given a desire to appear more knowledgeable and we have acknowledged this possibility. 

4. “Almost half of the general practitioners (41.2%) report residing in London and four-fifths are working full-time” – at this point lack of randomness is obvious but there is also another issue inevitably associated with such proportion of respondents. London is certainly not representative of the rest of the country. It is the opposite – the differences between London and the rest of UK are more than clear in almost every economic and social aspect. The authors did not mention about this issue at all and in my opinion it is crucial to at least highlight the possible differences.

Detail on the representativeness of the sample was included in S3. As noted by the reviewer, our sample of GPs used in the study is made up of 41.5% who reside in London. We relied on a reputable provider of GP samples (Panelbase) to recruit participants to our survey. Our results are robust despite the geographical representation because we have randomly assigned our respondents to the different treatments (attributes and levels) as per standard procedure for a discrete choice experiment. Thus, our groups are expected to be equivalent across conditions. 

5. “subsample of health professionals is predominantly comprised of white British (86.1%); geographically, they tend to be distributed across various regions of the United Kingdom.” - The authors should try to explain why there are such differences in geographical distribution between health professionals and GP.

We appreciate the concerns about the geographical distribution of our sample. Below (subgroup analysis) we examine whether this distribution has an impact on our results. Given that our paper is interested in explaining how the attributes of phage therapy condition health professional’s willingness to prescribe, we focus on any potential impacts on our results. 

6. “For example, 81.7% of the GP sample had previously heard of phage therapy, whereas this was only 17.5% for the general health professionals’ sample.” – this is exactly what I emphasized in my comment No. 4. GPs are better educated because half of them live in London. This geographical factor must be somehow addressed by the authors.

The following sentence has been added to the ‘Survey’ section (p.11):

“In addition to some possible overreporting of awareness, these differences are likely due to GPs having higher qualification and being more experienced in treating infections, prescribing medicines and thus being aware of the strategies for managing AMR.”

We have added a mention in the text that the respondents who are GPs tend to have higher qualifications, more experienced in prescribing medicines and are thus likely to have heard of this strategy. Also see our above point about overreporting of awareness as well. See our response to the ‘London impact’ below in point 7.

7. “However, the level of awareness about phage therapy differs drastically between GPs and other medical professions, with the former having a far greater awareness. This may reflect the difference in education syllabuses and day-to-day experiences of the two cohorts.” – such statement is not valid without analyzing health professionals who live in London. Education is only part of the problem. Again – London – is another important factor.

We have run a subgroup analysis of our experiment to test any London impact. The results are reported below. There are few differences. We note, in comparison to Fig 4 in the paper, that differences are due to differences between GPs and Health Professionals rather than to residing in London. Thus Fig 4 and its interpretation in the paper addresses this question raised by the review. 

Analysis by London vs other regions 1= London, 2 = Other Locations

8. “Another noticeable finding from the experiment is the consistently higher willingness of GPs to recommend phage treatment to patients, compared to general health professionals. The explanation for this could lie in the fact that GPs tend to have more hands-on experience in diagnosing and treating various medical conditions” – again – the "London" issue/impact.

We have added an explanation of this GP effect. Our subgroup analysis reported above shows minimal regional differences. Or put another way, any observed regional variation (London) is mostly driven by differences between GPs and Health Professional samples.

---

## [Decision Letter · Decision Letter 1]

19 Apr 2024

Factors of prescribing phage therapy among UK healthcare professionals: Evidence from conjoint experiment and interviews

PONE-D-23-33160R1

Dear Dr. Makarovs,

We’re pleased to inform you that your manuscript has been judged scientifically suitable for publication and will be formally accepted for publication once it meets all outstanding technical requirements.

Kind regards,

Adelaide Almeida

Academic Editor

PLOS ONE

Additional Editor Comments (optional):

Reviewers' comments:

Reviewer's Responses to Questions

**Comments to the Author**

1. If the authors have adequately addressed your comments raised in a previous round of review and you feel that this manuscript is now acceptable for publication, you may indicate that here to bypass the “Comments to the Author” section, enter your conflict of interest statement in the “Confidential to Editor” section, and submit your "Accept" recommendation.

Reviewer #2: All comments have been addressed

2. Is the manuscript technically sound, and do the data support the conclusions?

Reviewer #2: Yes

3. Has the statistical analysis been performed appropriately and rigorously? 

Reviewer #2: Yes

4. Have the authors made all data underlying the findings in their manuscript fully available?

Reviewer #2: Yes

5. Is the manuscript presented in an intelligible fashion and written in standard English?

Reviewer #2: Yes

6. Review Comments to the Author

Reviewer #2: All my concerns have been addressed. The paper is now suitable for publication. However, I still disagree with the sentence: "Phage therapy has been demonstrated as both safe and effective" as only clinical trials can confim the safety and/or efficacy of any therapy and somehow phage clincial trials have not been the most successful ones to date.

7. PLOS authors have the option to publish the peer review history of their article (what does this mean?). If published, this will include your full peer review and any attached files.

Reviewer #2: No
